# Health Disparities at the Intersection of Racism, Social Determinants of Health, and Downstream Biological Pathways

**DOI:** 10.3390/ijerph22050703

**Published:** 2025-04-29

**Authors:** Roland J. Thorpe, Marino A. Bruce, Tanganyika Wilder, Harlan P. Jones, Courtney Thomas Tobin, Keith C. Norris

**Affiliations:** 1Program for Research on Men’s Health, Hopkins Center for Health Disparities Solutions, Baltimore, MD 21205, USA; rthorpe@jhu.edu; 2Department of Health, Behavior and Society, Johns Hopkins Bloomberg School of Public Health, Baltimore, MD 21205, USA; 3Faith, Justice and Health Collaboratory, University of Houston Population Health, University of Houston, Houston, TX 77204, USA; mabruce@central.uh.edu; 4Department of Behavioral and Social Sciences, University of Houston, Tilman J. Fertitta Family College of Medicine, Houston, TX 77204, USA; 5School of Allied Health Sciences, Division of Health Sciences, Florida A&M University, Tallahassee, FL 32317, USA; tangywilder@gmail.com; 6Department of Microbiology, Immunology and Genetics, University of North Texas Health Science Center, Fort Worth, TX 76107, USA; harlan.jones@unthsc.edu; 7Institute for Health Disparities, University of North Texas Health Science Center, Fort Worth, TX 76107, USA; 8Department of Community Health Sciences, Fielding School of Public Health, University of California, Los Angeles, CA 90995, USA; courtneysthomas@ucla.edu; 9Division of General Internal Medicine and Health Services Research, David Geffen School of Medicine at UCLA, Los Angeles, CA 90024, USA

**Keywords:** social determinants of health, stress, race, racism, health disparities, stressors, biological mechanisms, theoretical frameworks

## Abstract

Despite overall improvements in the accessibility, quality, and outcomes of care in the U.S. health care system over the last 30 years, a large proportion of marginalized racial and ethnic minority (minoritized) groups continue to suffer from worse outcomes across most domains. Many of these health disparities are driven by inequities in access to and the scope of society’s health-affirming structural resources and opportunities commonly referred to as structural drivers or social determinants of health—SDoH. Persistently health-undermining factors in the social environment and the downstream effects of these inequities on neurocognitive and biological pathways exacerbate these disparities. The consequences of these circumstances manifest as behavioral, neurohormonal, immune, and inflammatory and oxidative stress responses, as well as epigenetic changes. We propose a theoretical model of the interdependent characteristics of inequities in the SDoH driven by race-based discriminatory laws, policies, and practices that eventually culminate in poor health outcomes. This model provides a framework for developing and validating multi-level interventions designed to target root causes, thereby lessening health disparities and accelerating improved health outcomes for minoritized groups.

## 1. Introduction

Health is defined by the World Health Organization as a state of complete physical, mental, and social well-being and not merely the absence of disease or infirmity [1]. The word health derives from the Anglo-Saxon root hal, meaning ‘to be whole’ [2]. In the United States, a nation founded on values of liberty, justice, and equality for all citizens, disparities in health across racial/ethnic groups remain a stark example of how far we are from fulfilling the seminal civil rights vision of the U.S. Constitution. However, U.S. history is interwoven with laws, practices, and policies driven by White supremacy ideology. This ideology of innate group-level hierarchies, based on a man-made socio-political creation of racial groups, led to the institutionalization of chattel slavery of Africans, the oppression and genocide of Native Americans, and the continuing marginalization of racial and ethnic minority (minoritized) groups instantiates a robust historical trend [3,4,5,6,7,8,9].

Science and medicine, as human enterprises, have not been immune to White supremacy ideology and have historically played a role in the justification of race-based discriminatory principles and practices, with the legacy effect of correlations between self-reported race/ethnicity and health outcomes, through the reification of race as a biologic construct. Recognizing that the field of medicine is in large part a social science, and not purely a natural science [1], diminishes the “biological authority” of the field and contributes to the exposure of White supremacist doctrine masquerading as biological claims. Indeed, much of health care focuses on sociodemographic factors. While pharmacologic treatments have an important role in medicine addressing lifestyle, a focus on behaviors and the structural drivers/social determinants of health (SDoH) remain paramount in practicing the art of medicine. The SDoH are summarized as conditions in the environments where people are born, live, learn, work, play, worship, and age. This includes, but is not limited to, socioeconomic status, access to quality health care and trust of medical institutions, education, health literacy, nutrition, built environment, and green space [10] reinforcing health and the field of medicine as more of a social than a natural science. The fallacies of science in service to racism have in fact prompted the medical community to re-evaluate how the life and health sciences define and operationalize race [11]. Race is a social construct that has been operationalized as a scheme of human classification through racism, a multi-layered system of structuring resources and opportunities, and assigning social value, based superficially on physical appearance and explicitly designed to disadvantage some individuals and communities and to advantage other individuals and communities [12]. Accordingly, the actual biological underpinnings of the social construct are largely driven by racism and its direct and indirect downstream impact, including stress on oppressed groups, expressed as weathering or biologic differences in allostatic load, inflammatory markers, and other biologic markers across man-made racial groups [13,14,15,16]. The biology of racism is often wrongly conflated with human biological classifications. The few clinically relevant differences in the prevalence of select gene polymorphisms/variants across groups defined and differentiated by physical appearance are grounded in ancestral geo-evolutionary pressures and founder effects and are not directly related to race or ethnicity.

The elimination of racial/ethnic health disparities and the improvement of overall population health have been at the center of the national public health agenda for almost 40 years. In 1985, the U.S. Department of Health and Human Services Secretary’s Task Force on Black and Minority Health (also known as The Heckler Report) documented stark differences in Black and White populations in health and health-related outcomes [17]. Much of the focus to reduce disparities following this report has been directed to the limited access to and/or receipt of low-quality health care for minority populations [18]. Yet despite emerging medical advances and improved access to care, many of these disparities still remain today [19], leading to a much broader question: Why? The persistence of health and health care disparities is owing, at least in part, to the nation’s inability to effectively deal with structural racism and its impact on the SDoH [20]. While U.S. federal agencies do not describe racism as a distinct social determinant of health, the World Health Organization does recognize structural racism as well as the socioeconomic and political systems that drive the inequitable distribution of power, money, and resources, as SDoH [21]. Several large systematic reviews have codified racism as an important contributor to poor mental and physical health [22,23,24]. The lack of societal will to address structural racism and inequities in the distribution of SDoH limits efforts to redirect and redistribute resources to communities in greatest need to remediate these persisting disparities [24]. This challenge is astutely summarized by Griffith and colleagues, who posit that health care disparities are the result of the intersection of a complex system (health care) and a complex problem (racism) [9].

Consideration of this complexity is critical to examining the interplay of social and biologic factors that lead to disparities, with the caveat that race and racism in America play an important role in the genesis and perpetuation of health and health care disparities not only at the individual level but, more importantly, at the policy level, which impacts the design, delivery, and financing of care [20]. The impact that structural racism and discrimination continue to have on individual- and community-level stress, mostly via their impact on SDoH, is likely underestimated. This underestimation is often derived from lack of specificity as to which of the many pathways link race and racism to a given health problem and to health disparities more broadly [9]. Stress is one common pathway that merits better understanding and that, when adequately addressed, may lead to substantive reductions in health disparities. The term stress is used in common parlance to indicate a state of exaggerated and/or prolonged exposure to adverse physical and/or psychologic assaults, which necessitate a physical/cellular response to maintain homeostasis [15]. When such exposures overwhelm an individual’s ability to effectively counteract them, they may lead to maladaptive behavioral and/or physiologic responses [15,25,26], the point when stress may be better termed distress. Discriminatory-based stress can also be aptly characterized as the cumulative impact of repeated experience with racialized social and/or economic adversity, negative SDoH, and political marginalization leading to a “wear and tear” on the body, known as weathering [13,14,27]. This often includes the activation of a number of neuro-hormonal and physiologic systems [14,15], inflammatory cytokines [16], stress-responsive signaling pathways [28], and more [29]. Here, we focus on a more explicit role of structural racism in the U.S., operating in large part through SDoH, within a conceptual biopsychosocial model of stress in racial/ethnic health disparities.

Although scholars have articulated several vexing problems underlying health disparities research [30], two major foci have received relatively little attention. Firstly, much of the research and intervention regarding health disparities has been intra-disciplinary and natural science oriented, leaving aspects such as the impact of racism limited to social science domains and rarely entering the more traditional biological or medical domains. Secondly, there remains a limit to our understanding of and attention to the effect of psychosocial and environmental stressors and their impact on health disparities through stress-induced biological mechanisms. While there is a substantial amount of research conducted under the label of racial/ethnic health disparities, much of it lacks the integration of the role of racism (especially structural/institutional racism) [9]. Structural/institutional racism operates at several levels, including through the SDoH (e.g., education, employment, access to care, environmental toxin exposures, criminal justice system) as well as downstream biologic effects (e.g., weathering, allostatic load, inflammation). Examination of these various levels and their complex interaction is needed to clarify how racism produces or exacerbates disparities in the incidence and/or complications of many medical disorders [9,31,32].

Several conceptual models have been developed by researchers to frame key aspects of health disparities, stress and health, and/or the interaction of race/racism and health [9,31,33,34]. Griffith et al. describe the theory behind dismantling racism to reduce health disparities [9], while Hill and colleagues, in providing a framework for investigators to think about the level of disparities-focused analyses [31,35], have developed a biopsychosocial model that requires an interdisciplinary approach. Myers [36] created a conceptual model that considers the interaction of ethnicity, socioeconomic status, and psychosocial adversities, including ethnicity-related stresses that result in biopsychosocial vulnerabilities, but it does not explicitly capture the role of structural, individualized, or even internalized racism. Bruce and colleagues have integrated and extended work by Seeman and Crimmins [35], Myers [36], and others [37,38] to demonstrate how sociological factors like racism create stressful social environments that shape physiological and behavioral responses adversely impacting health, well-being, and mortality, but also did not explicitly include racism in the actual model [39]. Methodological approaches that are consistent with these frameworks, coupled with considerations regarding life course perspectives and intersectional approaches, may prove to be valuable for scholars seeking to advance health disparities research, develop effective interventions that target root causes, and publish findings consistent with emerging recommendations for publishing standards on racial and ethnic health inequities [40].

While many of the previously mentioned frameworks seek to explicate how health disparities are produced, the focus on the role of racism and not race, as a primary genesis of stress in the pathway to racial/ethnic health disparities, is limited. As illustrated in Figure 1 and building on previous work [39], we have created a theoretical framework that emphasizes racism as a driving factor of disparate levels of inequities and stress in minoritized groups across multiple socio-ecologic levels that interact to contribute to racial/ethnic health disparities. We offer this model as a guide for researchers when considering how stress uniquely links an array of neurocognitive and biological pathways that may produce many of the major disparities in health outcomes. In brief, we posit that racial/ethnic disparities in health outcomes are a result of both interdependent and joint associations among factors, which we have classified into three domains: socioenvironmental, psychosocial, and behavioral factors. 

Socioenvironmental factors include key elements that impact the environment in which people work, live, and play. These factors range from ambient toxic exposures to constraints on health care access and resources that promote healthy living and encourage upward mobility, and all are heavily influenced by structural/institutionalized discrimination, residential segregation, and more [20,41]. Psychosocial factors relate to the impact that past or present psychological insults have on the mental and physical well-being of individuals and include such characteristics as anger, anxiety, chronic daily life stress, gender, and socioeconomic-related stress, often due to chronic exposure to structural/institutionalized racism [20,42,43]. Behavioral factors relate to actions that may arise as coping mechanisms to structural/institutionalized discriminatory exposures, with either negative impact on health outcomes such as smoking, physical inactivity, substance abuse, violence, and overeating and poor/inadequate nutrition, or resilience such as a deepening dependence on religion and spirituality [44]. Collectively or individually, the subsequent responses to each of the above influences entails activation of neurocognitive, neurohormonal, cellular oxidative, immune, and inflammatory signaling pathways, and possibly epigenetic changes that may adversely impact common chronic medical conditions [15,16,29,36,45]. Treatment of these conditions most often relies on the U.S. health care system, which (i) at present has little influence on SDoH, a major driving force in stress-induced health disparities; (ii) has created a chasm that diminishes trust and limits the willingness of minoritized groups to engage the health care system fully; and (iii) largely remains in denial or unaware of the role of racism and its biologic impact that may attenuate otherwise effective treatment modalities [46,47]. Thus, it is not surprising that racial/ethnic disparities in health persist.

We propose the inclusion of racism with an emphasis on structural/institutionalized racism as critical to the interconnectedness of the traditional stress-related socioenvironmental, psychosocial, and behavioral factors as well as downstream cellular pathophysiological mechanisms that underlie health disparities. Because minoritized groups are exposed to higher rates of structural/institutionalized social stressors than their White peers [48,49], they exhibit disparities in incidence, range, and severity of chronic medical conditions, neurocognitive and mental health outcomes, and overall rates of premature morbidity and mortality.

## 2. The Biology of Stress in Racial/Ethnic Disparities

Physiological stress is described by McEwen as the response to pressures placed on the body when one must cope, adapt, and/or adjust over extended periods of time [15]. Essentially, there are two broad types of stress: acute and chronic. Acute stress elicits a short-term array of neurohormonal responses, including the release of cortisol as well as catecholamines such as epinephrine and norepinephrine, in large part as a result of the activation of the hypothalamic–pituitary–adrenal (HPA) axis [15]. The acute stress process is generally an adaptive mechanism that is protective and positively contributes to survival. By contrast, exposure to seemingly nominal but continuously stressful situations, such as engendered by racism, may increase the heart rate, change blood pressure, and lead to the chronic activation of neurohormonal, cellular stress, and inflammatory pathways. These processes, albeit protective acutely, become maladaptive over the course of an individual’s life and contribute to alterations in the homeostatic processes, thereby accelerating the progression of many chronic diseases and premature senescence of cells/tissues/organs [15,50].

These factors that are driven by the unique history of racism in America add a large level of supplemental adversities that expose minoritized groups to excess stressors that “get under the skin” and disrupt normal physiologic processes, leading to an even higher physiologic burden or allostatic load [14,51,52,53]. Thus, the cumulative physiological burden exacted by the many stressors associated with adverse SDoH such as poverty and low education attainment is recognized as a risk factor for several highly prevalent chronic diseases [14,51,54]. The higher prevalence, earlier onset, and/or severity of many of these common chronic conditions among racial/ethnic minorities reinforces the linkage between the distribution or maldistribution of SDoH across communities leading to additional burdens of stress (e.g., financial exigency, neighborhood disadvantage, lack of access to care) and subsequent health disparities that occur at unknown point(s) along the course of their life [43].

## 3. Biologic Pathways

Research has shed light on not only how the effect of psychological stress may lead not only to dysregulation of neurocognitive pathways, such as altered executive function (reduced memory ability to implement tasks) and maladaptive behaviors with ultimately negative health consequences, but also how adverse health outcomes may be mediated through the neuroendocrine and other biologic systems [28,55,56,57,58]. Inflammation is an important pathway linking stress effects to health outcomes in the context of racial discrimination [59,60]. For example, stress-induced interactions between immune and neuroendocrine systems impact inflammatory processes (e.g., blood flow, capillary dilatation, leukocyte infiltration, and localized production of inflammatory mediators). The transcriptional activation of the antioxidant response element plays a crucial role in modulating oxidative stress and inflammatory responses in cells. Imbalance of this system may be a major mediator that leads to organ dysfunction, regardless of inherent biologically protective mechanisms [61]. Additionally, despite the 1.5–2-fold prevalence of several protective stress- and longevity-related gene polymorphisms such as GSTM1 (glutathione-S-transferase-m1) and the G allele of FOXO3 (rs2802292) found in some racial and ethnic minority groups [62,63,64], minoritized communities in America still suffer disproportionately from higher rates of many stress-related disease conditions and associated poor outcomes.

Specifically relating to stress and experiences of racial discrimination, Thames et al. recently reported that the increased activity of two key pro-inflammatory transcription control pathways (weathering factor κB [NF-κB] and activator protein 1 [AP-1]) and two stress-responsive signaling pathways (cAMP response element binding protein [CREB] and glucocorticoid receptor) were associated with differences in experiences of racial discrimination [16]. Chae et al. reported that experience of racial discrimination was associated with telomere shortening, a sign of premature aging [27]. Similarly, Gillespie and Anderson reported that an increasing frequency of exposure to racial discrimination led to a reduction in glucocorticoid sensitivity (i.e., increased production of cortisol that generates feelings of stress and anxiety) [65]. These findings highlight the important clinical impact that racial discrimination (interpersonal racism) may have on health and/or clinical disease states as well as the need for novel prevention and early intervention strategies while efforts to achieve social and racial justice continue.

Through this theoretical framework of racism and stress and its relation to racial/ethnic health disparities, as shown in Figure 1, it is now more apparent that until broad social changes lead to a relative level of equity, multi-level approaches that also favorably impact the effects of stress or distress at a cellular level are needed to attenuate the adverse health effects of excess stress or distress on individuals and communities at risk [28,66,67]. In addition, a comprehensive set of psycho-social interventions (from cognitive behavioral care to contemplative practices) [68] to supplement biologic interventions (e.g., antioxidants) can help to support the national efforts to reduce health disparities and to improve health outcomes for all Americans. Future research should seek to address racial/ethnic disparities in all health outcomes using an interdisciplinary team who can provide insights into how broad SDoH, psychological, and behavioral factors can impact physiologic and biological processes. One example might include understanding how racism of all forms operates in contexts that impact SDoH, leading to alterations at the molecular level.

## 4. Conclusions

Achieving health equity requires addressing historical and contemporary injustices as well as overcoming exposure to adverse SDoH (Table 1) and associated downstream neurocognitive and biologic factors to improve preventable health disparities and enhance health and health care for all persons (Figure 2). This can be accomplished by the continued pursuit of scientific discovery that can lead to health improvement and health policy.

## Figures and Tables

**Figure 1 ijerph-22-00703-f001:**
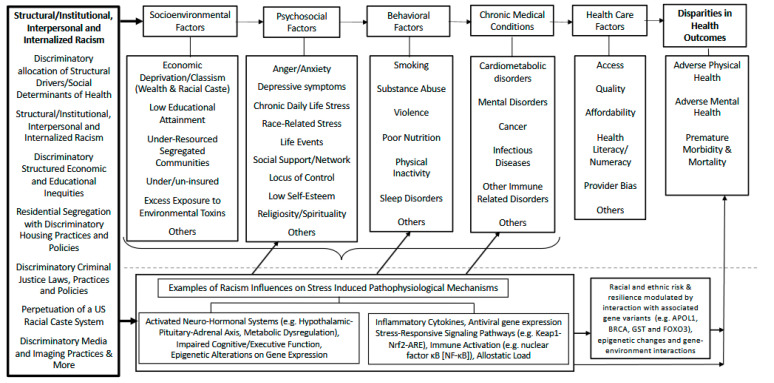
Theoretical Model of the associations between racism, SDoH, psychosocial, and behavioral factors, cellular stress systems, and health disparities. Legend: SDoH—social determinants of health; GST—glutathione S-Transferase; FOXO3—Forkhead box O3; ARE—antioxidant response element. Keap1—Kelch-like ECH-associated protein; Nrf2—NF-E2 p45-related factor 2; BRCA—BReast CAncer gene; APOL1—Apolipoprotein L1.

**Figure 2 ijerph-22-00703-f002:**
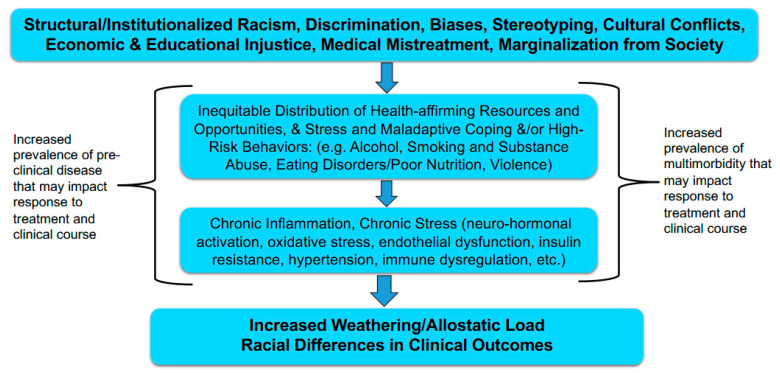
Associations between racism, social determinants of health, psychosocial, and behavioral factors, cellular stress systems, and health disparities. Adapted from [38,69].

**Table 1 ijerph-22-00703-t001:** Examples of different adverse social determinants of health associated with poor health outcomes and possible interventions.

Adverse Social Determinants of Health	Sample Mechanisms of Adverse Health Impact	Potential Interventions
Low Income	Low likelihood of any or quality health insurance Risk of being unhoused Risk of food insecurity Financial stress	Increase provider awareness of the impact of low income on health outcomes and potential patient resources such as social workers and community-based organizations Implement programs to address low income, income inequality, and living vs. minimumwage Unrestricted income supplements for low-income earners Housing vouchers for low-income earners
Limited or Absent Health Insurance and/or Access to Care	Reduced or inability to access preventive and early intervention Reduced or inability to access needed care for acute and chronic conditions Stress of uncovered health care and possible bankruptcy	Re-establish full funding for the Affordable Care Act (ACA) Navigator Program: provides free, one-on-one assistance to help people understand their ACA health insurance options Expansion of the ACA benefits Establish more federally qualified health centers (FQHCs) and expand early specialty care
Low Level of Educational Attainment	Limited health literacy Reduced job opportunities which can impact income and health insurance Stress of being disconnected from an increasingly digital society	Restructure public education to provide equal funding to all public schools rather than basing funding on community-level wealth which is inherently inequitable
Limited Health Literacy	Lesser ability to traverse a complex health care system Lesser ability to follow increasingly complex health recommendations Stress of being unable to navigate health at multiple levels	Implement age-specific health and wellness educational programs in K-12 to promote a national culture of health
Poor Nutrition	Increased risk of poor immune status Increased risk of obesity and cardiometabolic disorders Stress of premature cardiometabolic disease	Governmental incentives (e.g., tax credit) to food suppliers/retailers for providing affordable access to healthy foods De-incentivize availability to non-nutritious foods (e.g., fast foods) Incentivize food industry leaders to implement culturally and literacy tailored labeling Provision of healthy food to low-income, high-risk families
Limited Green Space Exposure	Increased risk of poor physical activity Increased risk of obesity and cardiometabolic disorders. Stress of premature cardiometabolic disease	Create more parks with walking spaces in the communities at highest risk Create activity spaces on school premises—support outdoor activities and reinstate physical education curriculum Health-informed traffic patterns and urban planning (e.g., to accommodate cyclists, sidewalks to promote walking) in high-risk communities
Distrust of Medical Institutions	Reticent to engage with health providers Reduced screening Delayed care Less likely to follow provider advice Stress of entering a system designed for health that has too often failed you, your family, and/or your community	Acknowledge historical injustices, such as unethical research practices, that have disproportionately targeted patients from racial and ethnic minority communities and make amends Ensure equity and ethical participation of underrepresented groups in clinical trials Revise media portrayal of sub-human narratives of racial and ethnic minority communities

## Data Availability

Not applicable.

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
