# Peer review of "Health Disparities at the Intersection of Racism, Social Determinants of Health, and Downstream Biological Pathways"

_ijerph, 2025, doi:10.3390/ijerph22050703_

Round 1

Reviewer 1 Report

Comments and Suggestions for Authors

Thank you for the opportunity to review the manuscript titled “Health disparities at the intersection of racism, social determinants of health, and downstream biological pathways.” The manuscript provides thorough historical and empirical evidence regarding the relationship between socio-economic and biological factors. The manuscript is also well-written and structured. The need for and the implications of the authors’ proposed theoretical model is also well argued. However, to further improve the utility and impact of the manuscript and the model, I have one substantive suggestion. There is a need for critical evidence or case study or specific potential research ideas to highlight the utility and impact of this theoretical model, specifically when compared to other existing models/theories, which also emphasize the relationship between socio-economic and biological factors. For example, authors have included the conceptual framework of weathering when outlining the existing models and theories in this area. However, the utility of the proposed theory will be clearly evident if authors can highlight limitations in the existing theories (for e.g., weathering) and how the proposed theoretical model addresses the gaps in the existing theories. Authors may employ evidence regarding lack of clear outcomes to highlight the utility of the proposed model. Authors may also highlight some clear examples of future research using the proposed model and how the consequent evidence can help better study and/or address the existing disparities. Having a more robust and clear discussion in this regard, especially after the theoretical model is introduced, will immensely help readers in appreciating the utility of the model. The clarity is especially important for those who may not be familiar with this line of research or are community members interested in engaging with scholars to improve health and well-being of their communities.

Author Response

Reviewer 1

Thank you for the opportunity to review the manuscript titled “Health disparities at the intersection of racism, social determinants of health, and downstream biological pathways.” The manuscript provides thorough historical and empirical evidence regarding the relationship between socioeconomic and biological factors. The manuscript is also well-written and structured. The need for and the implications of the authors’ proposed theoretical model is also well argued. However, to further improve the utility and impact of the manuscript and the model, I have one substantive suggestion. There is a need for critical evidence or case study or specific potential research ideas to highlight the utility and impact of this theoretical model, specifically when compared to other existing models/theories, which also emphasize the relationship between socio-economic and biological factors. For example, authors have included the conceptual framework of weathering when outlining the existing models and theories in this area. However, the utility of the proposed theory will be clearly evident if authors can highlight limitations in the existing theories (for e.g., weathering) and how the proposed theoretical model addresses the gaps in the existing theories.

Authors may employ evidence regarding lack of clear outcomes to highlight the utility of the proposed model. Authors may also highlight some clear examples of future research using the proposed model and how the consequent evidence can help better study and/or address the existing disparities. Having a more robust and clear discussion in this regard, especially after the theoretical model is introduced, will immensely help readers in appreciating the utility of the model. The clarity is especially important for those who may not be familiar with this line of research or are community members interested in engaging with scholars to improve health and well-being of their communities.

Thank you for this suggestion. Each of the theories that were articulated in the manuscript does provide an opportunity for researchers to consider broader contexts and how they impact racial/ethnic disparities. We do not view these as limitations, but rather our model explicitly calls our structural racism, SDOH, and stress as key drivers impacting racial/ethnic health disparities. Specifically, there is considerable attention to how mechanisms at the molecular level might be activated under broad social conditions. We have added a statement regarding future research with an example to inform the reader of the types of research that can be guided by this framework. We do see this framework as an opportunity for researchers to advance scientific inquiry by moving beyond studying racial/ethnic disparities in silos.

Reviewer 2 Report

Comments and Suggestions for Authors

My comments are included in the manuscript, overall the paper is strong and well-written and it feels like there are some gaps in the argument that are needed to create an argument with more continuity. 

Author Response

Reviewer 2

Page 2 “Recognizing that the field of medicine is in large part a social science”

I'm curious about this statement, as it's not widely recognized, which is the point I think is being made and therefore, could benefit from a sentence or two that helps make this argument more robust

Thank you for this recommendation. We now include additional sentences that add more clarity.  Specifically, we discuss the importance of the inclusion of SDOH to address racial/ethnic disparities in health. 

Page 3 “Here we focus on a more explicit role of U.S. racism within a conceptual biopsychosocial model of stress in racial/ethnic health disparities.”

These is a strong argument here and I'm wondering about the inclusion of stress as an example and wonder if a pathway linked to a SDoH would make more sense to maintain continuity of themes and create connections

Thank you for this suggestion. We have now included stress via SDoH as a pathway to understanding racial/ethnic health disparities. 

Page 3 “Griffith et al. describe the theory behind dismantling racism to reduce health dis-parities; while Hill and colleagues, in providing a framework for investigators to think about the level of disparities-focused analyses, have developed a biopsychosocial model that requires an interdisciplinary approach; while Myers has created a conceptual model that considers the interaction of ethnicity, socioeconomic status, and psychosocial adversities, including ethnicity-related stresses that result in biopsychosocial vulnerabilities. Bruce and colleagues have integrated and extended work by Seeman and Crimmins, Myers, and others to demonstrate how sociological factors like racism create stressful social environments that shape physiological and behavioral responses adversely impacting health, well-being, and mortality.”

Very long sentence, with the multiple models. It was a confusing read and took a few run through to grasp what is being shared, I would recommend this being broken up into different sentences and maybe a table that makes it easier to identify the models and their goals.

Thank you for this insight. We have broken this up into different sentences.  We agree it reads much better. 

Page 5 “The higher prevalence, earlier onset, and/or severity of many of these common chronic conditions among racial/ethnic minorities reinforce the linkage between the additional burden of stress and health disparities that occurs at unknown point(s) along the life course.”

I do see from this section that stress is meaningful for the argument and why it is included earlier in the paper; however, I still wonder about how this connects to the SDoH more specifically, I know race is not considered in SDoH, but community, poverty and so on are, which are all highly correlated.

Thank you for this insight. We have language to highlight the connection between stress, SDoH and racial/ethnic health disparities.

Page 6

Table 1. “Examples of different adverse social determinants of health associated with poor health outcomes and possible interventions.”

This table is really helpful and in your conclusion the reminder back to the downstream impacts is helpful. I just want to better understand how SDoH matters in this argument, they are not highlighted often and help create an understanding of the pathways to stress. My recommendation is just some added context to better connect these ideas to each other and bring the argument full circle.

Thank you for this suggestion. To add context we have added at column in Table 1 to highlight how select SDoH might mediate poor health and increase stress to better understand the many ways select SDoH can activate stress pathways.

Sincerely,

Keith C. Norris, M.D., Ph.D.

Distinguished Professor Dept of Medicine, David Geffen School of Medicine